# Physicochemical Composition and Sensory Quality of Goat Meat Burgers. Effect of Fat Source

**DOI:** 10.3390/foods10081824

**Published:** 2021-08-06

**Authors:** Alfredo Teixeira, Iasmin Ferreira, Etelvina Pereira, Lia Vasconcelos, Ana Leite, Sandra Rodrigues

**Affiliations:** 1Mountain Research Centre (CIMO), Escola Superior Agrária, Instituto Politécnico de Bragança, Campus Sta Apolónia Apt, 5300-253 Bragança, Portugal; ias.ferreira@hotmail.com (I.F.); lia.vasconcelos@ipb.pt (L.V.); anaisabel.leite@ipb.pt (A.L.); srodrigues@ipb.pt (S.R.); 2Escola Superior Agrária, Instituto Politécnico de Bragança, 5300-253 Bragança, Portugal; etelvina@ipb.pt

**Keywords:** burger, goat meat, olive oil, pork fat, oleogel, fatty acids, sensory

## Abstract

Several strategies for producing healthier meat products have been developed. Reducing fat content, using different fat sources, modifying and improving the fatty acid profile or even replacing saturated fat with oleogels are some of the methods used. Goat meat mainly from animals out of quality brands with low commercial value can be valorized when processed, giving the opportunity to increase its consumption and acceptability. Thus, the aim of this study was to study the effect of the replacement of pork as a source of fat with an olive oleogel in burgers manufactured with goat meat and to compare the goat meat burgers with the most common commercial burgers made with beef. Two replications of the burgers were manufactured at different times, and three samples of each burger type (GOO—goat meat burgers with olive oil; GPF—goat meat burgers with pork fat) were randomly selected from each lot manufactured. Each sample was analyzed in triplicate for each physicochemical analysis. At the time, the manufactured burgers were analyzed simultaneously with the commercial burgers. The burgers with olive oil (GOO) showed higher a* and b* than the burgers with pork fat (GPF) and consequently had lower h° and C*. The ashes, protein and collagen contents of the GOO and GPF burgers were similar to those of the other goat meat products. The effect of the incorporation of oleogel on the physicochemical composition of the burgers in relation to the pork fat was expressed in the fat content, 4 and 2.78% for GOO and GPF, respectively. CH burgers have significantly higher fat content (13.45%) than GOO and GPF burgers. The replacement of pork backfat with a vegetable oleogel modified the fatty acids profile, since the GOO burgers had the highest MUFA and PUFA and the lipidic quality, defined by the IA and IT indices, was 0.38 and 0.99, respectively. Globally, goat burgers were sensorially harder and presented a more difficult chewiness than CH. The replacement of the pork back fat with oleogel significantly decreased hardness and chewiness.

## 1. Introduction

Tendencies towards healthier ways of living among the global population have changed and are still changing several aspects of the food industry, leading producers to reformulate their products and offer more healthy alternatives [1]. Consumer perceptions towards healthier meat products are now mainly associated with how meat is produced and processed as well as its physicochemical composition and nutritional and sensory quality [2,3]. In recent years, several strategies for producing healthier meat products have been developed. Some of them go through reducing fat content [4,5] using different fat sources [6,7], modifying and improving the fatty acid profile [8,9,10] or replacing saturated fat with oleogels [11,12,13]. Most of the studies carried out are on different species of goats. In any case, consumers are increasingly valuing low-fat, high-quality products, and consequently there is increasing potential development of the goat meat market [14]. Goat meat is consumed commonly from younger animals less than 9 months old, as the flavor of the meat can be better perceived in older animals and often represents an undesirable characteristic for consumers, resulting in the disposal of the animals. Therefore, processed products could represent an opportunity to disseminate and increase the consumption of goat meat, particularly from older animals with low commercial value [15]. Many of the processed meat products, regardless of the type of meat, incorporate pork fat fractions, providing a representative amount of animal dietary fat, which is directly related to human health. The replacement of pork fat can consolidate the proposal to supply healthy goat meat products [16] to consumers with restrictions on the consumption of pork, such as Jews and Muslims, in addition to offering an option to consumers who seek the standards of a healthier diet, allowing a goat meat product to be inserted in different meat consumption markets [17]. In this sense, there is an increasing need for research in goat meat processing products, mainly in the control of the food processes, physicochemical characterization, food safety and sensory properties of new goat meat products, as pointed out by Teixeira et al. [5]. Therefore, the objective of this study was to study the effect of the replacement of pork as a source of fat with an olive oleogel in burgers manufactured with goat meat and to compare the goat meat burgers with a the most common commercial burgers made with beef.

## 2. Materials and Methods

### 2.1. Burger’s Production and Sampling

Burgers were prepared according to the Portuguese standard for meat and meat products [18], using fresh goat meat from the shoulder and loin carcass joints of adult (5 years old) Serrana Goats and fresh pork belly from the Bísaro breed. Burgers were produced at the Carcass and Meat Quality Laboratory at the Agriculture School of the Polytechnic Institute of Bragança following the flow chart fabrication shown in Figure 1.

Table 1 shows the formulations of two burgers produced at the laboratory. The composition of the traditional beef burgers (CH) acquired at a supermarket to compare with the goat burgers was: 90% beef, water, breadcrumbs (wheat flour, water and yeast) and starch (corn starch, antioxidants, sodium citrate, spices and the acidity regulator citric acid).

The oleogel incorporated olive oil used to produce the GOO burgers was prepared according to the guidelines set out by Barros et al. [19]: 56% of water, 6.7% of Prosella^®^, 6.7% of calcium sulphate, sodium alginate, wheat glucose, disodium diphosphate and sodium ascorbate and 37.3% of olive oil. The olive oil used was a *Trás-os-Montes* Protected Origin Designation (PDO) brand with the following fatty acids profile: 11.2% C16:0, 0.2% C17:1, 3.3% C18:0, 75.2% C18:1n-9, 7.7% C18:2n-6, 0.4% C20:0, 0.8% C18:3n-3, 0.2% C20:1n-9 and 0.1% C22:0 [20]. The pork fat incorporated was Bísaro pork belly with the following fatty acids profile: 1.3% C14:0, 22.3% C16:0, 2.1% C16:1, 11.9% C18:0, 41.9% C18:1n-9, 15.7% C18:2n-6, 1.2% C18:3n-3 and 0.78% C20:1n-9 [21].

Two replications of the burgers were manufactured at different times. Three samples of each burger type were randomly selected from each lot manufactured, and each sample was analyzed in triplicate for each physicochemical analysis. At the time, the manufactured burgers were analyzed simultaneously with the commercial burgers.

### 2.2. Physicochemical Analysis

The pH measurement was determined according to the Portuguese standard NP 3441 [22] using a Crison 507 pH-meter equipped with a 52–32 puncture electrode (Crison-instruments, Barcelona, Spain). Water activity was assessed according to AOAC [23] using a probe HigroPalmAw1 Rotronic 8303 (Bassersdorf, Switzerland). The hydroxyproline determination of the collagen content and concentration, protein, ashes and moisture were analyzed following the Portuguese standards NP 1987 [24], NP 1612 [25], NP 1615 [26] and NP 1614 [27], respectively. Total chloride content was analyzed according to the method specified in the Portuguese standard NP 1845 [28] expressed as sodium chloride as a percentage by mass.

Color was assessed using a Minolta CM-2006d spectrophotometer (Konica Minolta Holings, Inc., Osaka, Japan). The CIELAB space measured the coordinates L* (brightness), a* (redness) and b* (yellowness) as well as the color attributes C* (chroma) and h° (hue angle). 

The total pigment content [29], expressed as mg myoglobin/g fresh muscle, was obtained using the reflectance of the exposed surface by spectroscopy using a Spectronic Unicam 20 Genesys.

The total amount of lipids was extracted from 25 g of the burger sample according to the procedure employed by Folch et al. [30]. Fifty milligrams of fat were used to determine the fatty acid profile. The fatty acids were transesterified according to the method described by Shehata et al. [31] with the modifications by Domínguez et al. [32] and detailed in Teixeira et al. [20]. The results were expressed in g/100 g of fatty acids. Lipid quality was studied in terms of PUFA/SFA and n-6/n-3 ratios [33] as well as the index of atherogenicity (AI) and the index of thrombogenicity (TI) Equations (1) and (2), respectively, according to Ulbricht and Southgate [34]:(1)$$\mathrm{AI}=\frac{C12:0+4\times C14:0+C16:0}{\sum MUFA+\sum PUFA}$$
(2)$$\mathrm{TI}=\frac{C\;14:0+C16:0+C18:0}{0.5\times \sum MUFA+0.5\times \sum PUFA\;n-6+3\times \sum PUFA\;n-3+\frac{PUFA\;n-3}{PUFA\;n-6}}$$

### 2.3. Sensory Analysis

Samples of goat and beef burgers were sensorially evaluated by a taste panel. This panel was created after the recruitment, selection and training phases for the analysis of meat and meat products according to the Portuguese Standard (NP-ISO-8586-1, 2001) [35]. The chosen members of the panel were given specific training that allowed them to be prepared to evaluate the meat products in study. The whole process was conducted in the Sensory Analysis Laboratory at the Polytechnic Institute of Bragança.

The conditions of the test room where the evaluation took place followed standard guidelines (ISO-8589, 2007) [36]. The temperature was maintained between 20 and 22 °C and the relative humidity between 50 and 55%. The light in the room was white and each booth had a white light on to facilitate evaluation.

Three samples of each type of product were evaluated. Meat samples were wrapped in aluminum foil and cooked on thermal processing for 5 min for each side on an electric grill with heat above and below until the internal temperature center reached 70 °C (HI 935005, K-Type thermocouple thermometer, Washington, DC, USA).

Immediately after grilling, the samples were cut into 2 cm section pieces, wrapped in aluminum foil and placed in heaters to maintain the temperature of the samples.

The samples were randomly coded with three-digit numbers in monadic and random order. The panelists evaluated the samples according to the order established by the coordinator of the tests. They were informed of the need to clean their palate at the beginning and between the various samples of the session with mineral water and unsalted crackers.

The processed samples were evaluated in three sessions, three different samples per session. The panelists were required to observe, smell and taste all of the samples and give a judgment regarding the appearance (color intensity and brightness), odor (intensity and identification), oral texture (hardness, juiciness and chewability), flavor (basic taste, flavor intensity, identification and persistence) and acceptability of the product.

For evaluation, a 9 point scale was used with 1 representing the minimum (low intensity) and 9 the maximum (high intensity). Color intensity ranged from 1 (bright rose) to 9 (brown), brightness from 1 (none) to 9 (very bright), hardness from 1 (tender) to 9 (hard), juiciness from 1 (dry) to 9 (moist), chewability from 1 (easy) to 9 (difficult). The panelists also identified odor, basic taste and flavor from a list of possibilities. The samples were presented at random in each session. The methodology used was that described by the Standard (ISO-6658, 2005) [37].

### 2.4. Statistical Analysis

A Standard Least Square model was fitted to analyze the differences between the three types of burgers. The data were analyzed using the statistical package JMP^®^ Pro 16.0.0 by 2021 SAS Institute Inc. ©. The predicted means obtained were ranked based on pair-wise least significant differences and compared using the Tukey’s HSD test for * *p* < 0.05, ** *p* < 0.01 or *** *p* < 0.001 significance levels.

The statistical analysis of the sensory data was performed using the XLStat program (Addinsoft, New York, NY, USA), a Microsoft Office Excel add-in. A generalized procrustes analysis (GPA), which minimizes the differences between assessors, identifies agreement between them and summarizes the sets of 3-dimensional data, was used to develop a sensory profile for the burgers. Data matrices of 3 (types of hamburgers) by 8 (sensory attributes) for the 8 assessors were matched to find a consensus. The results are expressed in the form of graph that represents the respective sensory profiles in order to characterize them.

## 3. Results and Discussion

### 3.1. Physicochemical Properties and Chemical Composition 

Table 2 shows the results of the pH, a_w_ and color parameters. No statistical differences were found between the burgers for pH, with values varying between 5.44 and 5.97, although the goat meat burgers had a tendency towards slightly higher pH values. The burgers with pork fat (GPF) have significantly lower water activity (a_w_) than the burgers with olive oil (GOO), and the commercial burgers (CH) show an a_w_ between the goat meat burgers. In any case, all burgers showed the characteristics of a perishable product with a shelf life between 24 and 48 h and with a_w_ higher than 0.9 [4,6,38]. Significant differences were found for the color parameters between the goat burgers (GOO and GPF) and the CH burgers. The CH burgers showed higher values of redness (a*), yellowness (b*) and brightness (L*) and consequently higher chroma (C*) and lower tom (h°), as expected given the high pigment content of beef with greater oxidation after the blooming time of CIELAB measurements, as has been pointed out in several studies of raw beef burgers [11,39]. Between the goat burgers, no significant statistical differences were found for L*, but the burgers with olive oil (GOO) showed higher a* and b* than the burgers with pork fat (GPF) and consequently had lower h° and C*. The same effect on the color parameters of fat replacement by microencapsulated fish oil [38] or by oleogel rich in oleic acid [40,41,42] was observed in sausages.

The physicochemical composition of burgers is shown on Table 3. CH burgers are significantly different from GOO and GPF burgers for all the physicochemical parameters except for the ash content in relation to the GPF burger. Of all the results, the lowest protein and highest fat values of commercial burgers stand out in relation to those of the goat meat burgers. The GOO burgers had the highest ash content in relation to the GPF and CH burgers. As expected, and taking into account the previously mentioned CIELAB coordinates and color attributes, the CH burgers had a higher hem pigment content. The ashes, protein and collagen contents of the GOO and GPF burgers were similar to other goat meat products [4,6,7,15].

The effect of the incorporation of oleogel on the physicochemical composition of the burgers in relation to the pork fat was expressed in ashes and fat contents. Regarding the % of fat, GOO burgers have a significantly higher value than GOF burgers, 4.0% and 2.8%, respectively. CH burgers have significantly higher fat content (13.45%) than GOO and GPF burgers. In this sense, the analysis of the lipid quality of the burgers takes on a particularly interesting aspect. Table 4 shows the fatty acid profile of the burgers studied and the indices expressing the nutritional quality, the polyunsaturated/saturated fatty acids, the index of atherogenicity (IA) and the index of thrombogenicity (IT).

In an overall assessment of the results on Table 4, there are significant differences between the burgers GOO and GPF in relation to the commercial ones (CH). The CH burgers have a lipidic quality between the GOO and GPF burgers as can be verified by analyzing the P/S, IA and IT indices. The fatty acid profile of the three burgers is mainly composed of monounsaturated fat (oleic acid: C18:1n-9) and saturated fat (palmitic acid: C16:00). GOO burgers have significantly higher oleic content and lower palmitic than GPF and CH burgers. GPF shows the highest content of palmitic and stearic (C18:00) acids, while the GOO burgers have the lowest content of these fatty acids. The most representative polyunsaturated fatty acid is the linoleic acid (C18:2n-6), and the GOO burgers have a higher content of this fatty acid than the GPF and CH burgers. The proportions of these AG in the burgers are within the values found for goat meat [43,44,45,46] and goat meat products [4,6,7,15,47], as well as other meat products incorporating vegetable oleogels [12,13,19,48,49].

The GPF burgers show a significant higher content of elaidic acid (9t-C18:1) than the GOO and CH burgers. The content of this fatty acid in goat burgers is higher than in other goat meat products such as pâtés [6], goat meat sausages [4] or goat cured legs [15]. The elaidic acid used to be the main TFA isomer in industrial hydrogenation [47] and, like other *trans* fatty acids, consumption is related to coronary heart disease by several organizations of food nutrition. The total amount of TFA in meat ranging from 2–5% of the total fatty acid content [48] and the highest content was verified in GPF burgers (3.6%) below the cited range, since no other *trans* fatty acids were recorded. The content of elaidic acid in goat burgers was relatively higher than in other meat products with fat replacement by different oil emulsions [10,19,38,39,48]. However, in the present study, no other *trans* fatty acids were found. As expected, and in agreement with the cited studies, the replacement of pork backfat with a vegetable oleogel modified the fatty acids profile, since the GOO burgers had the highest MUFA and PUFA contents as well as the highest PUFA/MUFA ratio. Also, the pork back fat replacement with olive oleogel causes a clear reduction of the n-6/n-3 ratio from 18.55 to 7.26. As a consequence, the lipid quality of GOO burgers, defined by the IA and IT indices, was 0.38 and 0.99, respectively, showing superior quality to GPF burgers. The CH had intermediate IA and IT indices, better than GPF but worse than GOO. The same positive effect on IA and IT indices was verified in the studies of animal fat replacement by oleogels independently of the meat species [42,48,50,51].

### 3.2. Sensory Analysis

Eight sensory attributes were used to describe the differences between burger samples. Although training was given, variability among the panelists will always exist, and P3 and P4 were the ones with the higher residuals (Table 5). Panelists P1, P4, P7 and P8 used a wider part of scale because they have scaling factors higher than 1. GOO presented the higher residual of 3.26, followed by GPF with 2.62. CH was the most consensual, with a residual of 0.29. All panelists’ variability was mainly explained by F1, which agrees with 94% of the total variability explained by the same factor when applying a Generalized Procrustes Analysis.

To minimize the differences between the assessors, GPA was used to find a consensus (Figure 2). The biplot shows the consensus configuration with the correlations between sensory attributes, GPA factors F1 and F2 and the coordinates of the different burgers. Two factors explained all data variability; F1 and F2 together explained 100% of the total variability in hamburgers, where F1 explained 94.09% and F2 explained 5.91%. Previous research always needed at least three factors to explain all the variability of data, and the explained percentage was inferior to 100%. For example, 93% [51] in fresh goat meat, 88% [52] in fresh sheep and goat meat sausages, 73% [53] in fresh sheep meat and 79.49% in sheep and goat pâtés [54].

The coordinates of the different burgers and the correlation between the sensory attributes and the main factors led to indicate CH as brighter and juicier. It is known that a higher fat content can provide greater juiciness, softness and more intense brightnes in addition to contributing to a more pleasant flavor of the meat products. Studies [55,56] drew the same results, proving that fat interacts with other ingredients and affects the stabilization of meat emulsions, positively influencing the flavor and texture of the final product. 

Lower preferences for goat meat can be linked to the different goats’ collagen content and muscle fiber characteristics, both of which can affect tenderness [57].

Globally, goat burgers were harder and presented a more difficult chewiness than CH. The odor was what stood out the most in the pork fat burger. The olive oil burger presented a higher value for color, meaning it presented a darker color. The substitution of pork back fat by oleogel significantly decreased hardness and chewiness. A similar trend was reported [58] when 50% of the animal fat of burgers was replaced with oleogel from canola oil and when evaluating beef burgers with olive oil oleogel-based emulsion replacing animal fat [59]. In a study by Rodrigues et al. [51], olive oil had a positive influence on sheep and goat pâtés’ juiciness in relation to pork fat. The beef patty [60] formulation with 50% olive oil and 50% pork fat was considered easier to chew, soft, juicy and of a desired flavor when compared to a series of other formulations with different percentages of olive oil, canola oil and pork fat.

## 4. Conclusions

This study confirms the potential of oelogel technology to incorporate olive oil in goat meat burgers, providing a better lipidic quality. In sensory terms, the replacement of pork fat with olive oil produced a decrease in hardness and chewiness and a better flavor. 

## Figures and Tables

**Figure 1 foods-10-01824-f001:**
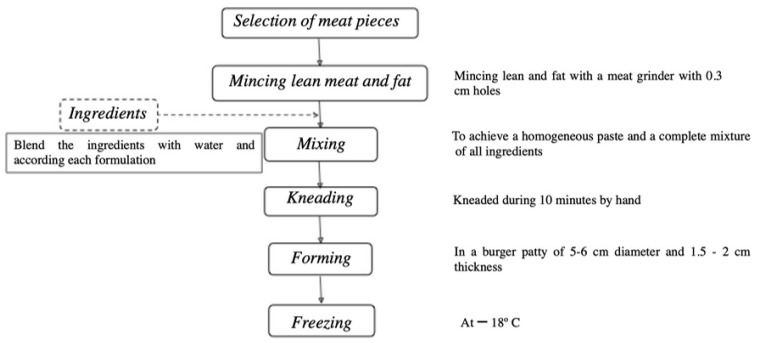
Flow chart of the fabrication of burgers.

**Figure 2 foods-10-01824-f002:**
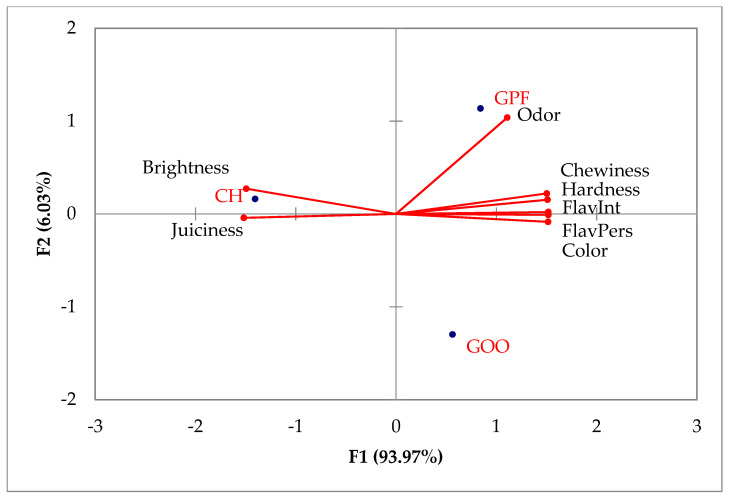
Consensus configuration: representation of the correlation between the sensory attributes F1 and F2 and the coordinates of the hamburgers.

**Table 1 foods-10-01824-t001:** Formulations of the two burgers manufactured.

Formulation	Burgers	
	GOO	GPF
Meat	87.9%	87.9%
Pork Fat	-	4%
Olive Oil Prosele	4%	-
NaCl	1.1%	1.1%
H_2_O	7%	7%

GOO—Goat meat + Olive oil; GPF—Goat meat + Pork fat.

**Table 2 foods-10-01824-t002:** Effects of fat replacement by olive oil on the pH, a_w_ and color parameters of goat burgers as compared with commercial beef burgers.

Parameter		Burgers		SEM	Significance
	GOO	GPF	CH		
**pH**	5.97	5.96	5.44	0.38	ns
**a_w_**	0.96 _a_	0.93 _c_	0.95 _b_	0.00	***
**Color**					
**L***	46.42 _b_	45.53 _b_	52.15 _a_	0.74	***
**a***	11.68 _b_	8.54 _c_	30.03 _a_	0.40	***
**b***	15.11 _b_	13.23 _c_	24.82 _a_	0.42	***
**h**°	52.25 _b_	57.20 _c_	39.58 _a_	0.35	***
**C***	3.66 _b_	3.29 _c_	5.23 _a_	0.05	***

SEM: Standard Error of the Mean; Significance: *** *p* < 0.001; ns: not significant; _a–c_, mean values in the same row (corresponding to the same parameter) not followed by a common letter differ significantly (*p* < 0.05; Tukey test).

**Table 3 foods-10-01824-t003:** Effects of fat replacement by olive oil on the physicochemical composition of goat burgers as compared with commercial beef burgers.

Parameter		Burgers		SEM	Significance
	GOO	GPF	CH		
Moisture (%)	74.23 _a_	74.48 _a_	63.36 _b_	1.00	***
Ashes (%)	1.82 _a_	1.23 _b_	1.46 _b_	0.10	**
Protein (%)	19.04 _a_	19.08 _a_	17.43 _b_	0.19	***
Fat (%)	4.0 _b_	2.78 _c_	13.45 _a_	0.25	***
Chlorides (%)	1.6 _a_	1.21 _a_	0.43 _b_	0.02	***
Collagen (%)	0.34 _b_	0.40 _b_	0.59 _a_	0.04	**
Pigments	0.47_b_	0.50 _b_	0.56 _a_	0.02	*

SEM: Standard Error of the Mean; Significance: * *p* < 0.05; ** *p* < 0.01; *** *p* < 0.001; ns: not significant; _a–c_ mean values in the same row (corresponding to the same parameter) not followed by a common letter differ significantly (*p* < 0.05; Tukey test).

**Table 4 foods-10-01824-t004:** Effects of fat replacement by olive oil on the fatty acid profile (expressed in g/100 g of fatty acids) of goat meat burgers as compared with commercial beef burgers.

Fatty Acids		Burgers		SEM	Significance
	GOO	GPF	CH		
C 10:00	0.05 _a_	0.03 _c_	0.07 _b_	0.00	***
C 12:00	0.05 _c_	0.06 _b_	0.08 _a_	0.00	***
C 14:00	1.24 _a_	2.94 _b_	1.85 _c_	0.03	***
C 14:1	0.06 _a_	0.59 _b_	0.09 _c_	0.00	***
C 15:0	0.32 _a_	0.40 _b_	0.44 _c_	0.00	***
C 15:1	0.03	nd	0.02	0.00	ns
C 16:0	19.56 _a_	27.23 _c_	23.25 _b_	0.05	***
C 16:1 n7	1.41 _c_	3.62 _a_	1.86 _b_	0.05	***
C 17:0	0.67 _b_	0.98 _a_	0.77 _ab_	0.08	*
C 17:1	0.59 _b_	0.56 _b_	0.70 _a_	0.01	***
C 18:0	13.26 _c_	18.78 _a_	17.79 _b_	0.15	***
9t-C 18:1	1.28 _c_	3.60 _a_	1.90 _b_	0.02	***
C 18:1 n9 c	54.38 _a_	46.48 _b_	37.30 _c_	0.18	***
C 18:2 n6 c	4.85 _a_	2.89 _b_	2.88 _b_	0.16	***
C 20:0	0.17 _a_	0.13 _b_	0.09 _c_	0.00	***
C 18:3 n6	0.01	nd	0.01	0.00	ns
C 20:1 n9	0.16 _b_	0.19 _a_	0.13 _c_	0.00	***
C 18:3 n3	0.45 _a_	0.16 _c_	0.33 _b_	0.01	***
C 21:0	0.28 _c_	0.33 _b_	0.40 _a_	0.00	***
C 20:2 n6	0.01	0.01	0.03	0.00	ns
C 20:3 n6	0.06	0.06	0.06	0.01	ns
C 20:3 n3	0.20 _a_	0.05 _b_	nd	0.00	***
C 20:4 n6	0.76 _a_	0.14 _b_	0.61 _a_	0.09	***
C 20:5 n3	0.06	nd	0.03	0.01	ns
C 22:6 n3	0.06	nd	0.06	0.00	ns
SFA	35.62 _c_	50.89 _a_	44.75 _b_	0.26	***
MUFA	57.91 _a_	45.85 _c_	51.18 _b_	0.20	***
PUFA	6.47 _a_	3.26 _c_	4.08 _b_	0.27	***
PUFA/SFA	0.18 _a_	0.06 _c_	0.09 _b_	0.01	***
n-6/n-3	7.26	18.55	7.70	0.38	***
IA	0.38 _c_	0.79 _a_	0.56 _b_	0.01	***
IT	0.99 _c_	1.96 _a_	1.48 _b_	0.01	***

nd—not detected; SFA: saturated fatty acids; MUFA: monounsaturated fatty acids; PUFA: polyunsaturated fatty acids; P/S: polyunsaturated/saturated fat ratio: IA index of atherogenicity; IT index thrombogenicity; SEM: Standard Error of the Mean; Significance: * *p* < 0.05; *** *p* < 0.001; ns: not significant; _a–c_ mean values in the same row (corresponding to the same parameter) not followed by a common letter differ significantly (*p* < 0.05; Tukey test).

**Table 5 foods-10-01824-t005:** Residual variance, scaling factors, and percentage variation explained by the first two principal components for each assessor for burger sensory analysis.

Panelist	Residuals	Scaling Factor	F1	F2
1	0.4966	1.2646	89.4235	10.5765
2	0.6774	0.7726	97.2779	2.7221
3	1.1843	0.9959	99.4956	0.5044
4	1.0446	1.0934	84.7405	15.2595
5	0.8226	0.8720	94.2082	5.7918
6	0.9427	0.7864	98.0826	1.9174
7	0.8814	1.8048	84.3475	15.6525
8	0.1180	1.2697	94.8560	5.1440

## Data Availability

Not applicable.

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
