# Peer review of "Physicochemical Composition and Sensory Quality of Goat Meat Burgers. Effect of Fat Source"

_foods, 2021, doi:10.3390/foods10081824_

Round 1

Reviewer 1 Report

The manuscript deals with a very topical issue, namely the upgrading of meat with low commercial value. The article reads well and is clearly written.

Abstract

The abstract contains the information it should. However, 2 small comments:

In the abstract, it is difficult to follow the difference between the abbreviations GFF and GPF. I suggest you keep them, but whenever you mention ‘burger’ in the abstract, make sure you clearly state 'beef burger' or 'goat burger'. Or should GFF be replaced by GPF?

Line 17: “GFF - goat meat bugers with pork fat”: add ‘R’: buRgers

Introduction

The introduction is short but contains enough information as an introduction to the rest of the manuscript.

M&M

Line 64-65: “…adult (5 years old) Serrana Goat breed and fresh pork belly form Bísaro breed.”: and what was the age of the goats of the Bisaro breed?

Figure 1: nice and useful flow chart!

Line 93-94: “At the time the manufactured burgers were analyzed simultaneously with the commercial burgers.”: you added detailed information about the composition of the GPF and GOO burgers, but nothing about the composition of the commercial burgers. Please add also this crucial information.

Line 124: “Samples of goat and beef burgers were sensory evaluated by a taste panel.”: Can you add some information about the number of the panel members, the number of training moments, the time between the first training moment and the moment of the first moment of evaluating the burgers…

Statistical analysis

Line 158: statis-tical: remove ‘-‘

Results and Discussion

The discussion contains the necessary references.

Line 173: Please include the full word of 'aw' in brackets.

Line 189-191: In the caption, state what aw, L*, a*,... stands for.

Conclusion

The conclusion is short but clear.

Author Response

Answers to Referee 1.

Authors are very grateful to your comments which have improved the quality of the manuscript.

Abstract

The abstract contains the information it should. However, 2 small comments:

In the abstract, it is difficult to follow the difference between the abbreviations GFF and GPF. I suggest you keep them, but whenever you mention ‘burger’ in the abstract, make sure you clearly state 'beef burger' or 'goat burger'. Or should GFF be replaced by GPF?

R: The abbreviations were clarified the GFF does not exist.

Line 17: “GFF - goat meat bugers with pork fat”: add ‘R’: buRgers

R: DONE

Introduction

The introduction is short but contains enough information as an introduction to the rest of the manuscript.

M&M

Line 64-65: “…adult (5 years old) Serrana Goat breed and fresh pork belly form Bísaro breed.”: and what was the age of the goats of the Bisaro breed?

R: All goat meat used was from goats 5 years old

Figure 1: nice and useful flow chart!

Line 93-94: “At the time the manufactured burgers were analyzed simultaneously with the commercial burgers.”: you added detailed information about the composition of the GPF and GOO burgers, but nothing about the composition of the commercial burgers. Please add also this crucial information.

R: The composition of commercial burgers is in lines 74 to 76

Line 124: “Samples of goat and beef burgers were sensory evaluated by a taste panel.”: Can you add some information about the number of the panel members, the number of training moments, the time between the first training moment and the moment of the first moment of evaluating the burgers…

R: This information was provided in lines 241 to 270 as well as the number of panel members in line 484.

Statistical analysis

Line 158: statis-tical: remove ‘-‘

R; OK

Results and Discussion

The discussion contains the necessary references.

Line 173: Please include the full word of 'aw' in brackets.

R: Done

Line 189-191: In the caption, state what aw, L*, a*,... stands for.

 R: Is in the lines 178 to 181

Conclusion

The conclusion is short but clear.

Reviewer 2 Report

This is an interesting manuscript and adds to our knowledge of how producing healthier meat products, reducing fat content, using different fat sources, modifying and improving the fatty acid profile. This paper has a practical and scientific value. . The results are interesting, properly explained and discussed. Some minor corrections are required, see below.

Lines 15 and 60: Change vulgar by common

Line 63: Add reference

Line 74: what muscles?

Lines 114-116: These indices are currently in question. The amount of PUFA n-3 is more relevant than the ratio n-6/n-3. The authors should read the following papers:

Harris, W.S. The Omega-6:Omega-3 ratio: A critical appraisal and possible successor. Prostaglandins, Leukotrienes and Essential Fatty Acids 2018, 132, 34–40, doi:10.1016/j.plefa.2018.03.003.

Harris, W.S.; Del Gobbo, L.; Tintle, N.L. The Omega-3 Index and relative risk for coronary heart disease mortality: Estimation from 10 cohort studies. Atherosclerosis 2017, 262, 51–54, doi:10.1016/j.atherosclerosis.2017.05.007.

Line 158: correct “statistical”

Line 174: I recommend to use CB as acronym for commercial burgers i plsce to HC (makes sense in Portuguese)

Compare 18:3 n-3 (table 4) with EFSA recommendations (Dietary Reference Values)

Table 4: Change n-6/n3 by n-6/n-3

Lines 244, 246 and 248 : “trans” must be in italic

Author Response

Referee 2

Authors are very grateful to your comments which have improved the quality of the manuscript.

This is an interesting manuscript and adds to our knowledge of how producing healthier meat products, reducing fat content, using different fat sources, modifying and improving the fatty acid profile. This paper has a practical and scientific value. . The results are interesting, properly explained and discussed. Some minor corrections are required, see below.

Lines 15 and 60: Change vulgar by common

  1. Done

Line 63: Add reference

R: Added

Line 74: what muscles?

R: The commercial product label did not indicate the names of the muscles.

Lines 114-116: These indices are currently in question. The amount of PUFA n-3 is more relevant than the ratio n-6/n-3. The authors should read the following papers:

Harris, W.S. The Omega-6:Omega-3 ratio: A critical appraisal and possible successor. Prostaglandins, Leukotrienes and Essential Fatty Acids 2018, 132, 34–40, doi:10.1016/j.plefa.2018.03.003.

Harris, W.S.; Del Gobbo, L.; Tintle, N.L. The Omega-3 Index and relative risk for coronary heart disease mortality: Estimation from 10 cohort studies. Atherosclerosis 2017, 262, 51–54, doi:10.1016/j.atherosclerosis.2017.05.007.

R: Thank you for your suggestion. In fact, there are different opinions in this matter, but this discussion falls out of the context of our study.

Line 158: correct “statistical”

R: Done

Line 174: I recommend to use CB as acronym for commercial burgers i plsce to HC (makes sense in Portuguese)

R: The abbreviation was uniformized to CH

Compare 18:3 n-3 (table 4) with EFSA recommendations (Dietary Reference Values)

R: It would be very interesting these comparisons, but in one other context of one other study.

Table 4: Change n-6/n3 by n-6/n-3

R: Changed

Lines 244, 246 and 248 : “trans” must be in italic

R: Done

Reviewer 3 Report

Review ID: foods-1312855

The reviewed article entitled ‘Physicochemical composition and sensory quality of goat meat burgers. Effect of fat source’ is quite interesting and contributes to update the literature data about the meat products from goat meat. Manuscript is really good organized. The experiment had been good planned and executed. Authors have obtained interesting results, which are presented generally, in a clear and transparent way. It is a pity that the authors did not include the results for fatty acids as content in mg/100 g which would be more valuable from a nutritional point of view.

Moreover, I have some major and minor objections to this work.

Main comments

I suggest to modify the title, for example: ‘Physicochemical properties and sensory attributes of goat meat burgers: effects of fat source’.

Introduction

  • L35-41: The authors here should refer to meat from slaughter animals in general, giving possible strategies (for instance Meat Science 2013, 95, 919–930, org/10.1016/j.meatsci.2013.03.030), rather than citing results for meat from sheep and goats (including some of their own) or new trends (for instance Appl. Sci. 2021, 11, 188, doi.org/10.3390/app11010188).

Material and Methods

  • Fig. 1 for subsequent stages I propose: Kneading, Forming and Freezing
  • Please, specify freezing and thawing conditions (method, temperature, time)
  • In Tab. 1 replace comma with dot, and express the values with the same accuracy
  • L72-75: Burgers are produced by many manufacturers, so what selection criteria were used for the control group. Were the control burgers subjected to freezing and thawing like the burgers from the experimental groups?

Results and Discussion

  • Table 3. In the M&M section, only the methodology for determining moisture and not dry matter is given, thus I strongly recommend removed results for dry matter, they are simply superfluous and not correct. The significance for moisture, as well values for fat and chlorides (GOO) should be corrected.
  • Why is the significance of differences in tables denoted in subscript?

Other remarks

L17, 21, 24,25: GFF or GPF, please unify here and throughout the text

L24: HC burgers group are not defined before

L29: HC or CH, please unify here and throughout the text

L40: please remove dot after [8-10]

L50-54: please redraft

L74: neat (meat) is redundant

L97: provide manufacturer for pH-meter

L98-99: … Rotronic 8303 (Bassersdorf, Switzerland).

L105:  … Osaka, Japan). Measurement results were expressed as CIE L*a*b coordinates, L* (lightness), … and h° (hue angle).

L107: please compare and correct the unit for pigments in table 3

L112: please specify the unit for fatty acids here and in table 4

L128: toke? Please correct (e.g. was conducted)

L134: unfrozen, please see note in section M&M

L138: replace cooking with grilling

L158: statistical

L161: please unify ‘P or p‘ throughout the text and tables

L171-172: physical replace with physicochemical or enumerate pH and aW. Please, comment on the lack of differences for pH (5.44 vs. 5.97 and 5.96)

L179 and Tab. 2: should be h° (hue angle) not h*

L212: Goo

L239: for elaidic acid I propose the following notation C18:1 trans-9

Author Response

Referee 3

Authors are very grateful to your comments which have improved the quality of the manuscript.

The reviewed article entitled ‘Physicochemical composition and sensory quality of goat meat burgers. Effect of fat source’ is quite interesting and contributes to update the literature data about the meat products from goat meat. Manuscript is really good organized. The experiment had been good planned and executed. Authors have obtained interesting results, which are presented generally, in a clear and transparent way. It is a pity that the authors did not include the results for fatty acids as content in mg/100 g which would be more valuable from a nutritional point of view.

Moreover, I have some major and minor objections to this work.

Main comments

I suggest to modify the title, for example: ‘Physicochemical properties and sensory attributes of goat meat burgers: effects of fat source’.

R: Thanks for your suggestion. In fact, this could be the title, but given that what is dealt with is the physicochemical composition, we do not understand that it should be changed composition for properties

Introduction

  • L35-41: The authors here should refer to meat from slaughter animals in general, giving possible strategies (for instance Meat Science 2013, 95, 919–930, org/10.1016/j.meatsci.2013.03.030), rather than citing results for meat from sheep and goats (including some of their own) or new trends (for instance Appl. Sci. 2021, 11, 188, doi.org/10.3390/app11010188).

R: Since this is a study on goat meat, we do not see that papers on meat in general should be cited instead of specific studies on sheep and goats.

Material and Methods

  • Fig. 1 for subsequent stages I propose: Kneading, Forming and Freezing

R: Done

  • Please, specify freezing and thawing conditions (method, temperature, time)

R: Was specified in the graphic

  • In Tab. 1 replace comma with dot, and express the values with the same accuracy

R: Done

  • L72-75: Burgers are produced by many manufacturers, so what selection criteria were used for the control group. Were the control burgers subjected to freezing and thawing like the burgers from the experimental groups?

R: There are commercial burgers and we did not control the process!

Results and Discussion

  • Table 3. In the M&M section, only the methodology for determining moisture and not dry matter is given, thus I strongly recommend removed results for dry matter, they are simply superfluous and not correct. The significance for moisture, as well values for fat and chlorides (GOO) should be corrected.

R: Ashes correspond to reference 26 NP-ISO-1615/2002. Dry Matter results were removed. The significance of moisture was corrected, Th results of fat and chlorides are corrected.

  • Why is the significance of differences in tables denoted in subscript?

R: We do not understand the question!

Other remarks

L17, 21, 24,25: GFF or GPF, please unify here and throughout the text

R: The text and abbreviations were clarified

L24: HC burgers group are not defined before

R: Yes, they are defined on lines 77 to 79.

L29: HC or CH, please unify here and throughout the text

R: The abbreviation was uniformized as CH

L40: please remove dot after [8-10]

R: Done

L50-54: please redraft

R: The text was redraft

L74: neat (meat) is redundant

R: neat was removed

L97: provide manufacturer for pH-meter

R: Provided

L98-99: … Rotronic 8303 (Bassersdorf, Switzerland).

R: OK

L105:  … Osaka, Japan). Measurement results were expressed as CIE L*a*b coordinates, L* (lightness), … and h° (hue angle).

R: OK

L107: please compare and correct the unit for pigments in table 3

R: OK

L112: please specify the unit for fatty acids here and in table 4

R: Done

L128: toke? Please correct (e.g. was conducted)

R: Done

L134: unfrozen, please see note in section M&M

R: OK

L138: replace cooking with grilling

R: OK

L158: statistical

L161: please unify ‘P or p‘ throughout the text and tables

R: OK

L171-172: physical replace with physicochemical or enumerate pH and aW. Please, comment on the lack of differences for pH (5.44 vs. 5.97 and 5.96)

R: Done

L179 and Tab. 2: should be h° (hue angle) not h*

R: OK

L212: Goo

R: OK

L239: for elaidic acid I propose the following notation C18:1 trans-9

R: The notation used was (9t-C18:1)
